# IllusionCAPTCHA: A CAPTCHA based on visual illusion

## Abstract

CAPTCHAs have long been essential tools for protecting applications from automated bots. Initially designed as simple questions to distinguish humans from bots, they have become increasingly complex to keep pace with the proliferation of CAPTCHA-cracking techniques employed by malicious actors. However, with the advent of advanced large language models (LLMs), the effectiveness of existing CAPTCHAs is now being undermined.

To address this issue, we have conducted an empirical study to evaluate the performance of multimodal LLMs in solving CAPTCHAs and to assess how many attempts human users typically need to pass them. Our findings reveal that while LLMs can solve most CAPTCHAs, they struggle with those requiring complex reasoning—a type of CAPTCHA that also presents significant challenges for human users. Interestingly, our user study shows that the majority of human participants requir a second attempt to pass these reasoning CAPTCHAs, a finding not reported in previous research.

Based on the findings from our empirical study, we introduce IllusionCAPTCHA, an innovative approach designed to be "Human-Easy but AI-Hard". This new CAPTCHA employs visual illusions to create tasks that are intuitive for humans but highly confusing for AI models. Furthermore, we developed a structured, step-by-step method that to generate misleading options, which particularly guide LLMs towards making incorrect choices and reduce their chances of successfully solving CAPTCHAs. Our evaluation shows that IllusionCAPTCHA can effectively deceive LLMs 100% of the time. Moreover, our structured design significantly increases the likelihood of AI errors when attempting to solve these challenges. Results from our user study indicate that 86.95% of participants successfully passed the CAPTCHA on their first attempt, outperforming other CAPTCHA systems.

## CCS Concepts

• **Do Not Use This Code → Generate the Correct Terms for Your Paper**; *Generate the Correct Terms for Your Paper*; Generate the Correct Terms for Your Paper; Generate the Correct Terms for Your Paper.

## Keywords

CAPTCHA, Visual Illusion, Large Language Model

**ACM Reference Format:**

. 2018. IllusionCAPTCHA: A CAPTCHA based on visual illusion. In *Proceedings of Make sure to enter the correct conference title from your rights confirmation emai (Conference acronym 'XX).* ACM, New York, NY, USA, 9 pages. https://doi.org/XXXXXXX.XXXXXXX

## 1 Introduction

CAPTCHAs (Completely Automated Public Turing tests to tell Computers and Humans Apart) [24] are tools designed to distinguish human users from automated bots on websites. They ingeniously capitalize on humans' unique cognitive abilities, assigning tasks that are simple for people but difficult for machines. These tasks typically involve recognizing distorted text [24], selecting specific images [11, 15], or identifying patterns—activities that rely on human perception and intuition. Therefore, traditinoal CAPTCHAs can be categorized into two types: text-based and image-based. CAPTCHAs leverage the gap between human cognitive skills and the current limitations of AI, making them an essential tool in online security and ensuring the integrity of web interactions.

In the era of AI, techniques bring posibilities of automated CAPTCHA solving [16, 23, 28]. Early CAPTCHAs, such as text-based and image-based challenges, rely on tasks of text recognition and basic image identification which challenges the unique visual and cognitive abilities of humans. However, modern deep-learning models can now easily solve these types of challenges [21]. In response to this, reasoning-based CAPTCHAs [9] that requires more logical reasoning and common sense emerges. More recently, the evolution of Large Language Models (LLMs) has brought substantial improvements in both reasoning capabilities and multimodal processing [1, 22]. This technique has been applied to the development of new approaches to tackle reasoning-based CAPTCHAs [6]. However, there remains a gap in research: no study has systematically investigated the performance of multimodal LLMs across the full spectrum of CAPTCHA types.

In this paper, we first investigate the performance of multimodal LLMs on the task of CATPCHA solving. We evaluate two state-of-the-art models, GPT-4o [4] and Gemini 1.5 pro 2.0 [22], across different types of CAPTCHAs (e.g., text-based, image-based, and reasoning-based CAPTCHAs). We employ Zero-Shot prompting [17] and the Chain-of-Thought (CoT) prompting [27] as our primary methodologies. Additionally, we conducted a user study to assess how many attempts human users typically need to successfully pass these CAPTCHAs, which has not been considered in any papers before [6, 18].

The results of our investigation reveal four key insights: (1) LLMs perform better on text-based CAPTCHAs compared to image-based and reasoning-based CAPTCHAs. (2) While LLMs struggle with complex reasoning CAPTCHAs, their performance improves when using the CoT prompting, suggesting that with reasoning chains, LLMs have the potential to solve such challenges. This indicates that current CAPTCHAs may no longer be as secure as intended. (3) Our user study shows that although reasoning-based CAPTCHAs are difficult for AI to solve, they are also challenging for human users. These challenges can even frustrate users, diminishing their patience during attempts. (4) Finally, our study reveals that human users often make the same mistakes as LLMs, underscoring the

need to develop methods that can effectively differentiate between LLMs and human users.

The empirical study prompts us to explore the design of a new CAPTCHA that is AI-hard and human-easy. Specifically, we aim to create CAPTCHAs that more effectively differentiate between humans and bots. To this end, we propose IllusionCAPTCHA, which employs images embedded with visual illusions [30] — challenging for AI models to interpret, but easier for humans to perceive. These illusions take advantage of the human brain's unique ability to process visual and cognitive discrepancies, a capability that AI struggles to replicate. Additionally, to further improve the distinction between human users and bots, we incorporate a step-by-step question structure that prompts bots to make predictable errors. This design ensures that human users can easily pass these CAPTCHAs, while bots are more likely to fail by making consistent, recognizable mistakes.

The efficiency of IllusionCAPTCHA was evaluated using two advanced multimodal LLMs (GPT-4o and Gemini 1.5 pro 2.0). Through experiments, we find that these LLMs are unable to successfully pass our CAPTCHA implementation, and the step-by-step question structure effectively tricks them. Furthermore, the user study reveals that human participants are able to solve the CAPTCHA on their first attempt. These findings demonstrate that Illusion-CAPTCHA offers a higher level of security compared to traditional challenges. Additionally, it is easier for humans to solve than reasoning-based CAPTCHAs, while still maintaining a robust defense against AI models.

To summarize, we make the following contributions:

- We conducted a systematic empirical study to investigate the effectiveness of LLMs on CAPTCHAs and found that current CAPTCHAs are no longer secure. Furthermore, our user study reveals that, in most circumstances, users are unable to pass the current CAPTCHAs on their first attempt. To the best of our knowledge, this is the first study that systematically surveys LLM effectiveness on CAPTCHAs.
- We introduce IllusionCAPTCHA, the first illusion-based CAPTCHA that leverages the unique ability of the human brain to process visual information. Additionally, our step-by-step questioning approach effectively encourages bots to make predictable mistakes.
- We evaluate our method using two state-of-the-art models, GPT-4o and Gemini 1.5 pro 2.0. The experimental results demonstrate that our strategy effectively presents challenges for AI models to solve the generated CAPTCHAs, rendering it AI-hard, while simultaneously remaining accessible and straightforward for human users to navigate. This dual capability ensures that our CAPTCHA not only enhances security against automated attacks but also provides a user-friendly experience, bridging the gap between robust security measures and usability.

**Ethical Considerations.** We emphasize that our research and experiments on LLMs' effectiveness in solving CAPTCHAs have not been used for any unethical purposes or financial gain, and the user study we designed raises no ethical concerns. Unlike many studies focused on developing CAPTCHA solvers, our proposed

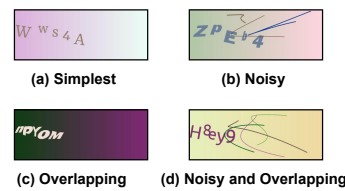

**(a) Simplest**  **(b) Noisy**

**(c) Overlapping**  **(d) Noisy and Overlapping**

**Figure 1: Text-based CAPTCHA**

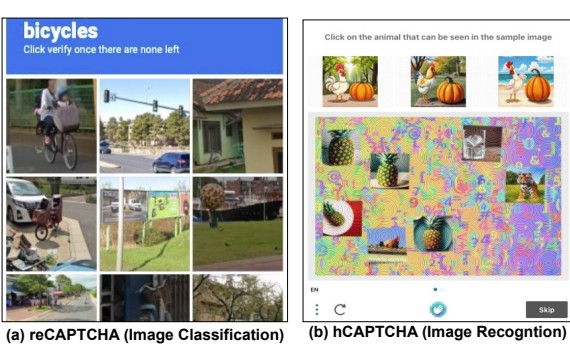

**(a) reCAPTCHA (Image Classification)**  **(b) hCAPTCHA (Image Recogntion)**

**Figure 2: Image-based CAPTCHA**

IllusionCAPTCHA aims to enhance web security by effectively defending against modern LLM-based CAPTCHA attacks.

## 2 Background

### 2.1 CAPTCHAs and CAPTCHA Solver

CAPTCHAs [6, 9] have evolved from simple text recognition to complex reasoning challenges to distinguish between human users and bots. This ongoing development mirrors the "cat-and-mouse" dynamic in cybersecurity, where both CAPTCHAs and CAPTCHA solvers become increasingly innovative in response to one another. This transformation has accelerated the shift from traditional CAPTCHA-solving (e.g. OCR[28]) methods to modern AI technology, posing a significant threat to the effectiveness of CAPTCHAs.

**Text-based CAPTCHAs.** Text-based CAPTCHAs are the earliest form of CAPTCHA, designed to leverage text recognition tasks that are easy for humans but challenging for machines. As shown in Figure 1(a), the simplest text-based CAPTCHAs consist of a string of English characters with no added noise. However, as machine learning techniques have advanced, text-based CAPTCHAs have become increasingly complex, incorporating more than just English characters and moving beyond simple backgrounds [7, 18]. However, the complexity of CAPTCHA also makes human users hard to identify.

**Image-based CAPTCHAs.** Image-based CAPTCHAs are the most popular CAPTCHAs used online. Compared to text-based CAPTCHAs, image-based CAPTCHAs needs more vision capture ability, with more abundant image categories in image content. Based on the particular workloads embeded in the image-based CAPTCHAs, we categorize them into two groups.

- **Object Classification.** This type of CAPTCHA (e.g., re-CAPTCHA [10], as shown in Figure 2(a)) typically presents a set of images and asks users to identify specific ones from various given categories. Early image-based CAPTCHAs relying on object recognition used relatively simple images. However, to combat increasingly sophisticated automated bots, modern image-based CAPTCHAs now incorporate noise and other distortions into the images, making it more challenging for AI systems to accurately recognize the objects. This added complexity aims to disrupt the efficiency of automated classification while still allowing human users to complete the task with ease.
- **Object Recognition.** Compared to object classification, object recognition demands a deeper level of visual understanding. For instance, hCAPTCHA [13] requires users to click on the correct images based on a given description, as shown in Figure 2(b). This task involves not only identifying objects but also understanding the context of the question and selecting images that match the description. Unlike simple object classification, which may only involve labeling objects in an image, object recognition in CAPTCHAs requires users to interpret complex scenarios or differentiate between visually similar objects.

**Reasoning-based CAPTCHAs.** The evolution of reasoning-based CAPTCHAs [26] signifies a shift from traditional visual recognition tasks to cognitive challenges that demand more advanced logical reasoning and image comprehension. As shown in Figure 3, reasoning-based CAPTCHAs usually need human users to click move some icons to pass the check. This development highlights the limitations of conventional CAPTCHA solvers(e.g. OCR) in handling these more complex tasks. However, reasoning-based CAPTCHAs also require users to engage in higher-level reasoning, which can lead to increased frustration and impatience among human users.

## 2.2 Large Language Models

The evolution of Large Language Models (LLMs) has transformed traditional AI learning method [1]. By increasing the scale of training data, model can significantly improve their ability to understand, generate, and process human language with greater accuracy and contextual relevance. Notably, recent advancements in multimodal LLMs [3, 4] have facilitated the integration of text and images, enabling AI systems to analyze complex visuals and describe them using natural language. While the reasoning capabilities of LLMs are still being evaluated, their potential to address reasoning-based tasks is both promising and continuously expanding [20]. Consequently, the capabilities demonstrated by LLMs pose a substantial threat to the security of traditional CAPTCHA systems [6].

## 2.3 Visual Illusion

Visual illusions [12, 14, 25] illustrate the complexities of human visual reasoning, demonstrating that our brain interprets the world in ways far more intricate than what we directly perceive. These illusions provide valuable insights into how cognitive processes, shaped by perception and context, influence our understanding of reality. While existing research shows that modern LLMs can

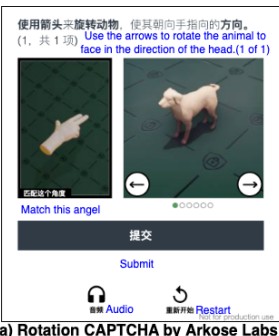
**(a) Rotation CAPTCHA by Arkose Labs**

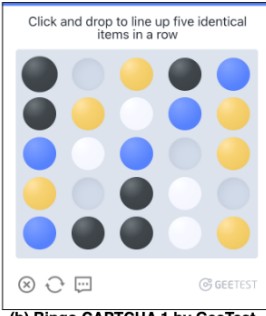
**(b) Bingo CAPTCHA 1 by GeeTest**

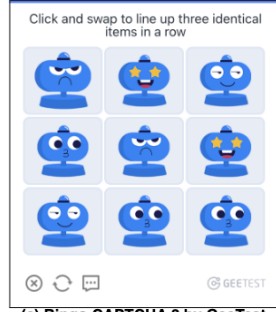
**(c) Bingo CAPTCHA 2 by GeeTest**

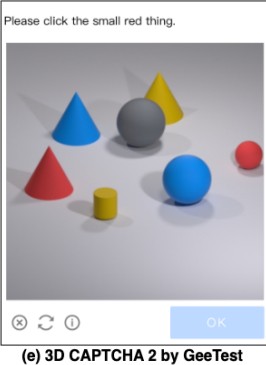
**(d) 3D CAPTCHA 1 by YiDun**        **(e) 3D CAPTCHA 2 by GeeTest**

**Figure 3: Reasoning-based CAPTCHA**

identify objects similarly to humans, their imaginative capabilities remain limited [30], making it difficult for them to match human-level reasoning.

## 3 Threat Model

In this paper, we outline our assumptions regarding the goals and capabilities of attackers.

**Attacker goals.** We assume that the adversary aims to automatically solve CAPTCHAs without human interactions, which could potentially lead to these results [5, 19]: (1) Automating Actions: Gaining unauthorized access to websites, applications, or services to automate tasks (e.g. account creation, data scraping, or spamming). (2) Credential Harvesting: Exploiting CAPTCHA weaknesses to gain access to user accounts by defeating login protections. (3) Fraudulent Activities: Engaging in malicious activities like ticket

scalping, purchasing limited-edition items, or bypassing purchase limits imposed by websites. (4) Disruption of Services: Creating bot networks that can flood websites with traffic, bypassing CAPTCHAs to disrupt normal operations.

**Attacker capabilities.** We assume the attacker is restricted to interacting with the CAPTCHA through the graphical interface, without using techniques such as reverse engineering, JavaScript decompiling, or direct code analysis. In this work, we primarily consider that the adversary abuses the capabilities of multimodal LLMs with their reasoning capabilities and object recognition capabilities. These LLMs can be utilized not only to solve CAPTCHAs but also to automate the entire attack process—from selecting target websites to registering accounts—enabling a highly efficient and scalable attack pipeline.

## 4 Empirical Study

We first conduct a systematic empirical study to assess the effectiveness of LLMs in identifying both traditional and modern CAPTCHAs. The full potential of LLMs in this area remains largely unexplored. Additionally, to address the knowledge gap among human users regarding CAPTCHAs, we design a user study to evaluate their performance when passing different CAPTCHA challenges. This investigation is structured around two research questions:

- **RQ1 (Effectiveness):** How effective are LLMs in accurately solving CAPTCHAs, and what types of errors are they most likely to make?
- **RQ2 (User Study):** Can human users properly solve different types of CATPCHA challenges? What are the difficulties in this process?

In the following of this section, we address the two research questions through two sets of experiments.

### 4.1 Effectiveness of LLMs in Solving CAPTCHAs

**CAPTCHA Categorization.** Different from other works [6, 18], we cover all categories of visual CAPTCHAs. Within each category, there are different designs from different vendors. Therefore, we collect the state-of-the-art commercialized CAPTCHAs that are available online, and summarize the detailed sub-categories as below.

- **Text-based CAPTCHAs.** We collect different types of text-based CAPTCHAs that requires users to recognize a series of letters or characters. After survey, we conclude four types of them available now. (1) **Simplest Text-based CAPTCHAs**, shown in Figure 1(a), is the simplest text-based CAPTCHAs, which is also the most popular CAPTCHAs online. This type of challenge can be solved easily by traditional CAPTCHA solvers. These typically feature clear, unaltered text, making them vulnerable to basic image recognition techniques. (2) **Noisy Text-based CAPTCHAs**, shown in Figure 1(b), introduce visual noise, such as random lines, dots, or distortions into the text CAPTCHA, which can interfere with traditional CAPTCHA solvers. Despite the added complexity, they still primarily ensure that users could recognize the contents within. (3) **Overlapping Text-based CAPTCHAs**,

shown in Figure 1(c), are a type of text-based CAPTCHAs that involve texts where characters are overlapped with each other at different angles. While this writing style is totally recognizable to humans, it is hard for traditional solvers [] that relies on segmentation strategies to solve. (4) **Noise-enhanced Overlapping Text-based CAPTCHAs**, shown in Figure 1(d), are the type of challenges combine both visual noise and overlapping texts, which significantly increases the difficulty for traditional CAPTCHA solvers to counter.

- **Image-based CAPTCHAs.** In addition to the traditional text-based CAPTCHAs, more recent ones include images that tests the common sense of users as a type of challenge. We conclude two types of basic image-based CAPTCHAs. (1) **reCAPTCHA** presents users with tasks like selecting images (image classification) that contain specific objects, such as traffic lights or crosswalks, or verifying street signs. Vastly adopted by Google, it is the most common types of CAPTCHA that has been well researched. There are three versions of reCAPTCHAs, with similar image patterns but different underlying mechanisms to counter traditional automated solutions such as JavaScript reverse engineering. (2) **hCAPTCHA** hCAPTCHA involves more detailed image recognition tasks, requiring users to have a stronger ability to understand the prompts (e.g., selecting images that contain wheels).

- **Reasoning-based CAPTCHAs** are new emerging category of challenges that aims to counter the automated solvers powered by deep learning methods. After survey, we identify three types of reasoning-based CAPTCHAs. (1) **Rotation CAPTCHAs**, also known as Angular by their developers, require users to adjust an object's orientation to align with a reference object. As shown in Figure 3(a), users need to properly recognize the orientation of two different objects (the finger and the lamb in this example) to solve the challenge. There are two versions of Rotation CAPTCHAs available in the market now, both devleoped by Arkose Labs. (2) **Bingo CAPTCHAs (Gobang & IconCrush)** is a new type of reasoning-based CAPTCHA also developed by Arkose Labs. As seen in Figure 3(b), this type of challenge tasks users with identifying and rearranging elements on a board to create a line of matching items. The types of elements and the rules for manipulation can differ widely based on the provider. For instance, in Figure 3(b), users can swap any two items without restriction, while in Figure 3(c), swaps are limited to adjacent items, illustrating the range of variation in this type of CAPTCHA. (3) **3D Logical CAPTCHAs**, as demonstrated in Figure 3(d) and Figure 3(e), requires users to choose an object from a 3D environment. This process requires users to identify the logical relationships tied to attributes like shape, color, and orientation of the objects within the challenge. For instance, in Figure 3(d), users must identify the number 0 that aligns with the orientation of a yellow letter W, whereas Figure 3(e) asks users to select the larger object positioned to the left of a green object.

**Table 1: Experimental results of applying the multi-model LLMs over the selected CAPTCHAs.**

| Method | | Zero-Shot | | COT | |
|---|---|---|---|---|---|
| Metric | | Success Rate | | Success Rate | |
| Model | | GPT4o-latest | Gemini 1.5 pro 2.0 | GPT4o-latest | Gemini 1.5 pro 2.0 |
| Text-based CAPTCHA | Simplest | 76.66% | 73.33% | 90.00% | 83.33% |
| | Overloaping | 66.66% | 60% | 70.00% | 60.00% |
| | Noise | 70.00% | 73.33% | 73.33% | 66.66% |
| | Noise+Overloaping | 36.66% | 23.33% | 50.00% | 43.33% |
| Image-based CAPTCHA | reCAPTCHA | 40.00% | 33.33% | 50.00% | 23.33% |
| | hCAPTCHA | 40.00% | 36.66% | 43.33% | 30.00% |
| Reasoning CAPTCHA | Angular | 13.33% | 0.00% | 13.33% | 0.00% |
| | Gobang | 0.00% | 0.00% | 6.66% | 3.33% |
| | IconCrush | 0.00% | 0.00% | 16.66% | 10.00% |
| | Space | 46.66% | 26.66% | 53.33% | 26.66% |
| | Space Reasoning | 33.33% | 20.00% | 40.00% | 23.33% |
| Average | | **38.48%** | **31.51%** | **46.06%** | **33.63%** |

**Table 2: Experimental results of applying the multi-model LLMs over the selected CAPTCHAs.**

| Attempt Times | First Attempt | Second Attempt | Third Attempt | More-time Attempt |
|---|---|---|---|---|
| Text-based CAPTCHA | 47.82% | 39.13% | 8.69% | 4.34% |
| Image-based CAPTCHA | 30.43% | 56.52% | 4.34% | 8.69% |
| Reasoning CAPTCHA | 21.73% | 43.47% | 21.73% | 13.04% |
| Average | **33.33%** | **46.37%** | **11.59%** | **8.69%** |

**Dataset Collection.** To rigorously assess the ability of LLMs to solve CAPTCHAs, we include the three types of CAPTCHAs as discussed in the Background: text-based, image-based, and reasoning-based CAPTCHAs. Notably, we exclude audio CAPTCHAs due to their limited usage online [8], which is mainly for visually impaired persons. Additionally, our study emphasizes real-world scenarios, so all the CAPTCHAs used were collected from website applications. Consequently, we built a dataset comprising three types of CAPTCHAs(text-based CAPTCHAs 1, image-based CAPTCHAs 2 and reasoning-based CAPTCHAs 3).

**Methodology.** To evaluate these CAPTCHAs, we employ two powerful LLMs (Gemini 1.5 pro 2.0 and GPT4-o) using both Zero-Shot and Chain-of-Thought (COT) methodologies. Each CAPTCHA category presents a unique set of challenges that necessitates specialized solving strategies. As a result, we utilize different prompts for the LLMs to predict outcomes for various CAPTCHAs, measuring success rates as our primary metric. We manually analyze each LLM response to ensure the accuracy of the results. In the zero-shot approach, a solution is considered correct only if the LLM outlines the exact procedure for solving the CAPTCHA. In contrast, in the CoT approach, a sub-step is deemed successful if the LLM's proposed solution for that specific sub-step is accurate.

**Result Analysis.** Table 1 presents the results of our evaluation of LLMs' effectiveness in solving CAPTCHAs. Using a zero-shot approach, LLMs can solve most text-based CAPTCHAs, with the exception of those featuring overlapping characters or significant noise. However, their accuracy drops to only 40% for image-based CAPTCHAs. Additionally, LLMs encounter challenges with reasoning-based CAPTCHAs due to their limited reasoning capabilities. Nonetheless, employing COT prompting significantly enhances the effectiveness of LLMs in identifying these types of CAPTCHAs. This

underscores the growing threat that advancements in LLMs pose to web security, indicating that current CAPTCHA methods may no longer be sufficiently secure.

**Answer to RQ1:** *Our verification experiment reveals that (1) LLMs perform better on text-based CAPTCHAs compared to image-based and reasoning-based CAPTCHAs; and (2) although LLMs struggle with complex reasoning CAPTCHAs, their performance significantly improves when employing the Chain-of-Thought (CoT) strategy. This suggests that with reasoning chains, LLMs have the potential to overcome these challenges. Consequently, this indicates that current CAPTCHAs may no longer be as secure as intended.*

## 4.2 User Study

To investigate human user behavior, we designed a user study in the form of a questionnaire. Since some CAPTCHAs cannot be repeatedly obtained from their original sources, we develop the user study by extracting CAPTCHA images from the their original web applications, and we manually label the correct answers for them. All images collected are sourced from the dataset used in our previous analysis.

**User Study Settings.** Our questionnaire allocates each participant 1 minute to solve a CAPTCHA. If they cannot complete it within that time, they must attempt it again until they succeed. During this process, we record all the successful and failed attempts. In the end, we have 23 human participants in our study.

**Result Analysis.** Table 2 presents the results of our user study, indicating that most participants are unable to solve the CAPTCHA on their first attempt. It is particularly challenging for users to identify image-based and reasoning-based CAPTCHAs, as evidenced by

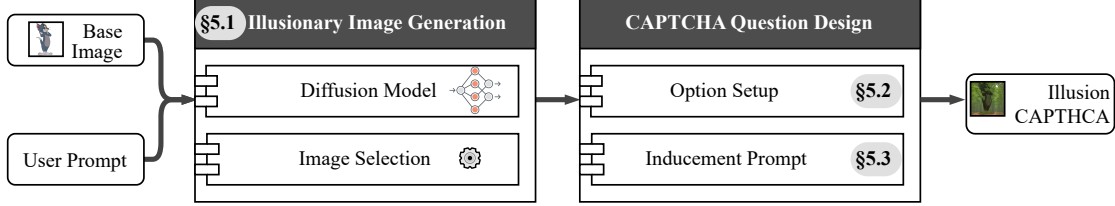

**Figure 4: Overview of IllusionCAPTCHA**

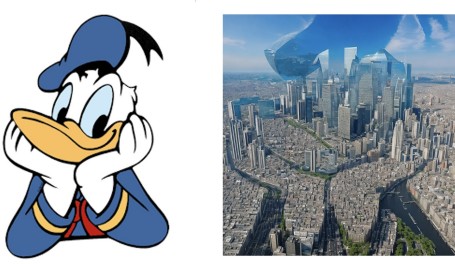

**Figure 5: An example of the original and illusionary image**

some individuals needing more than five attempts to successfully pass them.

**Answer to RQ2:** *The result of our user study reveals that (1) Our user study indicates that while reasoning-based CAPTCHAs pose significant challenges for AI, they are also difficult for human users. Consequently, these CAPTCHAs can easily frustrate users, leading to diminished patience during their attempts. (2) Furthermore, our study reveals that human users frequently make the same mistakes as LLMs, highlighting the need to develop methods that can effectively distinguish between LLMs and human users.*

## 5 Methodology

As illustrated in Figure 4, IllusionCAPTCHA generates CAPTCHA challenges through a three-step process. First, it blends a base image with a user-defined prompt, such as "huge forest," to create a visual illusion that obscures the original content. This results in images that, while recognizable to humans, can confuse AI systems. Second, multiple-choice options are generated based on the altered images, forming the CAPTCHA challenge options. Our empirical study indicates that humans may occasionally make errors similar to those of LLMs, suggesting that relying solely on illusionary images may not be sufficient to distinguish human users from bots. Therefore, we incoperate the third step of "Inducement Prompt" to induce our LLM-based attackers to choose the intended choice. Moreover, we utilize multimodel question to increase difficulty for attackers but easy for human users to identify. Below we detail the design of IllusionCAPTCHA.

## 5.1 Illusionary Image Generation

The first objective is to create illusionary images that are easily recognizable by humans but difficult for AI systems to identify. This process involves tackling two primary challenges: (1) maintaining the context of the base image, and (2) add disturbance to the image particularly effective for AI systems to interfere with their capabilities.

To address the first challenge, we employ an illusion diffusion model [2], which generates images by blending two different types of content. Built upon ControlNet [29], a framework that allows precise control over image generation through conditional inputs, this model ensures that the resulting images remain accessible to human viewers while being challenging for automated systems to interpret. Figure 5 shows how a normal image is transferred into an illusionary one. However, not all generated images will effectively balance recognizability for humans while fooling AI vision. To overcome the second challenge, we firstly generate 50 sample images using different seeds at a fixed level of illusion strength—an optimal number for human identification. We then calculate the similarity between each generated image and the base image, selecting the one with the lowest similarity, which can be seen as the most diffcult images for bots to identify.

To enhance the perceptibility of the generated images, we develop tailored strategies for two types of illusion-based CAPTCHAs: traditional text-based CAPTCHAs and image-based CAPTCHAs. In the first scenario, the base image contains a clear, readable word embedded within an illusion. To ensure that human users can still recognize the text with minimal effort, we opt for simple, familiar English words such as "day" or "sun". In the second scenario, the base image features a well-known, easily recognizable character or object, such as an iconic symbol or a famous figure from contemporary culture (e.g., "Mickey Mouse" or "Eiffel Tower"). This ensures that human users can quickly identify the content, even with added illusionary elements. These strategies aim to strike a balance between maintaining human usability and introducing complexity that misleads AI-based attackers.

## 5.2 Options Setup

Our options are meticulously designed to defend against attacks on LLMs. In our CAPTCHA, we offer four distinct choices. One option represents the correct answer, while another is the input sentence we utilize in our models. The remaining two options consist of detailed descriptions of our illusionary images, intentionally crafted without referencing any content from our true answer.

Unlike traditional CAPTCHAs that require users to type text or select multiple images to answer a question, our CAPTCHA asks users to choose the correct description of an image. This design simplifies the process by offering a hint, making it easier for users to identify the correct answer without needing to click through multiple images.

Compared to text-based CAPTCHAs, ours is more user-friendly, as it avoids the challenges posed by visual illusions. Additionally, in contrast to hCAPTCHA and reCAPTCHA, our approach reduces the difficulty of making a selection. Unlike reasoning-based CAPTCHAs that require users to manipulate images, which can lead to frustration, our design eliminates the need for such interactions, further improving user experience.

## 5.3 Inducement Prompt

Building on the our empirical study, we discover that both LLMs and human users tend to make similar errors when presented with certain types of CAPTCHAs. Additionally, human users often require a second attempt to pass the CAPTCHA successfully. As a result, relying on a single question to differentiate between AI and human users proves insufficient. To address this issue, we designed a system that aims to lure potential attackers, such as multimodal LLMs, into selecting predictable, bot-like answers. Our CAPTCHA format uses multiple-choice questions, each offering four answer options.

Our strategy centers on the idea to trick the LLM-based adversary to select the option that describes the illusionary element added, which is the object that LLMs typically fails to capture. Research [30] has shown that LLMs typically describe images with long, detailed sentences. To exploit this, we include one option that features an intentionally elaborate, detailed description of the illusionary elements in the image (e.g., "a vast forest filled with birds, depicting a beautiful and serene scene").

Additionally, to reduce the difficulty for human users, we embed hints within the questions that guide them toward the correct answer. These hints are crafted to trigger hallucinations in LLMs, further increasing the likelihood that bots will select incorrect responses.

## 6 Evaluation

To assess the performance of our IllusionCAPTCHA, we have structured our evaluation around three key research questions:

- **RQ1: (Human Identification of Illusionary Images)** Can the illusionary images generated by our solution remain identifiable to human users?
- **RQ2: (LLM Deception by Illusionary Images)** Can the illusionary images effectively deceive LLMs to select the false answer?
- **RQ3: (Effectiveness of Inducement Prompts)** Can the CAPTCHA structure we design compel bots to make targeted choices?
- **RQ4: (Human Attempts to Pass CAPTCHA)** How many attempts do human users need to pass our designed CAPTCHA?

**Table 3: Experimental results of RQ1**

| Metric | Visibility | Confidence |
|---|---|---|
| **Illusionary Text** | 83.00% | 4.80 |
| **Illusionary Image** | 88.00% | 4.90 |

## 6.1 RQ1: Human Identification of Illusionary Images

**Motivation.** In this section, we investigate whether illusionary images can effectively convey information to human users. This is a crucial step, as a CAPTCHA image must successfully communicate information to its intended audience.

**Method.** To address RQ1, we designed a questionnaire for human users to assess their ability to identify illusionary images. Our questionnaire includes two types of images: text-based illusionary images and image-based illusionary images, each containing five samples. Below are the details of our questionnaire.

- **Perception of Illusion (Mandatory Question):** "Do you notice any illusionary effect in this image?"
- **Uncertainty Clarification (Optional Question):** "If you are uncertain, could you please explain why?"
- **Confidence Level (Mandatory Question):** "If you answered 'Yes' or 'No' regarding the perception of an illusion, how confident are you in your response? Please rate on a scale from 1 (least confident) to 5 (most confident)."
- **Image Description (Mandatory Question):** "What do you observe in this image?"
- **Description Confidence (Mandatory Question):** "How confident are you in your description of the image? Rate from 1 (least confident) to 5 (most confident)."

**Result Analysis.** The key results from this survey are summarized in Table 3, 10 participants taking part in this questionnaire. In terms of visibility, the data reveals that human users were able to accurately identify 83% of illusionary text and 88% of illusionary images on average. This suggests a relatively strong ability to recognize deceptive or distorted content in both formats.

Additionally, the confidence metric provides insight into the users' perception of their own performance. The majority of participants reported high levels of confidence in their selections, indicating that they believed they were making correct judgments, even when faced with illusionary or complex content. This confidence may play a crucial role in how users engage with tasks that involve visual and textual interpretation, highlighting the special structure of human vision.

## 6.2 RQ2: LLM Deception by Illusionary Images

**Motivation.** In this section, we investigate whether illusionary images can effectively deceive the vision of LLMs. This is crucial, as a CAPTCHA image must successfully fool automatic bots.

**Method.** To rigorously test our generated illusionary images, we utilize the same settings as our empirical study in Section 4. In contrast to our empirical study, this section aims to demonstrate that LLMs cannot identify illusionary images. Additionally, unlike other studies, we require precise answers; for example, the answer should be "Jack mice" rather than simply "mice."

**Table 4: Experimental results of RQ2**

| Method | Zero-Shot | | COT | |
|---|---|---|---|---|
| Metric | Success Rate | | Success Rate | |
| Model | GPT4o-latest | Gemini 1.5 pro 2.0 | GPT4o-latest | Gemini 1.5 pro 2.0 |
| Inducement Prompt-The First Attempt | 100.00% | 100.00% | 100.00% | 100.00% |
| Inducement Prompt-The Second Attempt | 100.00% | 100.00% | 100.00% | 100.00% |

**Table 5: Experimental results of RQ4**

| Attempt Times | First Attempt | Second Attempt | Third Attempt | More-time Attempt |
|---|---|---|---|---|
| IllusionCAPTCHA | 86.95% | 8.69% | 0.00% | 4.34% |

**Table 6: Experimental results of RQ2**

| Method | Zero-Shot | | COT | |
|---|---|---|---|---|
| Metric | Success Rate | | Success Rate | |
| Model | GPT4o-latest | Gemini 1.5 pro 2.0 | GPT4o-latest | Gemini 1.5 pro 2.0 |
| Illusionary Text | 0.00% | 0.00% | 0.00% | 0.00% |
| Illusionary Image | 0.00% | 0.00% | 0.00% | 0.00% |

**Result Analysis.** Table 6 presents the experimental results of large language models (LLMs) in identifying illusionary images and texts. Our findings indicate that, in both Zero-Shot and COT reasoning, neither GPT nor Gemini successfully identified the illusionary images, with a 0% success rate. Interestingly, we observed that when using COT, GPT could recognize the shape of a character hidden in the image but was unable to accurately name the character, even when provided with a hint in the prompt. Therefore, visual illusion is very hard for current LLMs to identify, indicating it is a natural CAPTCHA.

## 6.3 RQ3: Effectiveness of Inducement Prompts

**Motivation.** In this section, we investigate whether our inducement prompts can lead our intended attackers (GPT-4o and Gemini 1.5 pro 2.0) to select the options we designed.

**Method.** In this experiment, we use GPT-4 and Gemini 1.5 Pro 2.0 as our LLMs. We apply two prompt settings: Zero-Shot and COT. Additionally, we allow the LLMs two attempts to identify the images, considering their ability to retain context across interactions.

**Result Analysis.** From Table 4, we can see that in both attempts, the LLMs consistently selected the option we predicted they would choose, suggesting that the models were identifying only the generated content and not focusing on what we intended human users to recognize. Additionally, we observed that the LLMs often selected the longest description of the images, indicating a tendency to overlook the core elements of the visual illusion.

This behavior highlights a key limitation in the LLMs' ability to process visual context effectively, as they appear to prioritize the length or complexity of the descriptions rather than engaging with the nuanced visual details. This finding suggests that while LLMs perform well with textual analysis, they may struggle when tasked with interpreting visual content that requires deeper contextual understanding or inference, such as illusionary images.

## 6.4 RQ4: Human Attempts to Pass CAPTCHA

**Motivation.** One of the primary aims of our CAPTCHA is to facilitate easier identification of images by human users. Therefore, it is crucial to demonstrate that our CAPTCHA is more user-friendly. To achieve this, we need to assess the number of attempts required for human users to successfully pass the CAPTCHA.

**Method.** In this experiment, we designed a questionnaire structure similar to the one used in Section 4 to investigate how many attempts human users need to pass our IllusionCAPTCHA.

**Result Analysis.** Table 5 presents the experimental results of our IllusionCAPTCHA for human users. In this survey, we consulted 23 participants, and we found that 86.95% were able to pass the CAPTCHA on their first attempt, while 8.69% succeeded on their second attempt. We also collected feedback on the reasons for failure and discovered that the primary reason participants could not pass was that they did not know the name of the character, although they recognized it as a character from television. Therefore, our CAPTCHA is more friendly for human users to identify, compared to current existing CAPTCHAs.

## 7 Conclusion

In this paper, we conduct an empirical study to assess the performance of LLMs in solving existing CAPTCHAs. Following this, we design a user study to determine how many attempts human users need to pass these CAPTCHAs. Based on the findings from our empirical study, we introduce IllusionCAPTCHA, aimed at facilitating the distinction between human users and automated bots. Our comprehensive evaluation demonstrates the effectiveness of IllusionCAPTCHA in generating images deceptive to automated solutions. The experimental results show that it presents significant challenges for AI models, while simultaneously remaining accessible and user-friendly for human users. This dual capability ensures that our CAPTCHA not only enhances security against automated attacks but also provides a seamless user experience.

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
