# OpenReview forum: "IllusionCAPTCHA: A CAPTCHA based on visual illusion"
_ACM.org/TheWebConf/2025/Conference — WWW 2025 Poster_

### Official Review · Reviewer_HsSP · 2024-11-04

**Novelty:** 3
**Technical Quality:** 4

**Review:**

Brief Introduction
The paper presents IllusionCAPTCHA, a new CAPTCHA approach based on visual illusions designed to enhance online security by making it "Human-Easy but AI-Hard." Traditional CAPTCHAs, particularly text and image-based ones, are increasingly vulnerable to attacks by advanced AI, especially multimodal large language models (LLMs). IllusionCAPTCHA leverages the human brain's ability to perceive visual illusions in a way that confuses AI. The method employs illusion-based images that are easy for humans to recognize but difficult for AI models to interpret, along with a step-by-step question format that induces predictable errors in AI responses.

Evaluation
 - Strengths:

1.The paper introduces a approach by applying visual illusions specifically to CAPTCHA design. This is novel in that it shifts from traditional methods (text recognition, object classification) to using cognitive illusions that exploit gaps between humans and AI.

2.The paper demonstrates sound technical implementation, including systematic empirical studies of various CAPTCHA types, user studies, and detailed evaluation of the proposed method's effectiveness in challenging AI.

 - Weaknesses

1.The concept of using visual illusions is not entirely new in computer vision; illusions have been studied extensively in adversarial AI contexts. Thus, the combination of these techniques, e.g. diffusion model, reduced innovation.

2.The paper's organization is not good. The Threat Model section is short and insufficiently integrated with the rest of the paper, while the Empirical Study and Evaluation sections are separated by a Methodology section, which disrupts the logical flow and makes readers confused.

3. Poor generalization performance: As AI’s capabilities in image recognition continue to improve, future advancements may lead to optimization methods specifically targeting visual illusions. This could gradually reduce the "AI-hard" characteristic of this CAPTCHA, resulting in diminished security. So how to improve generalization performance is an important aspect.

**Questions:**

See Evaluation above.

**Reviewer Confidence:**

3: The reviewer is confident but not certain that the evaluation is correct

**Scope:**

3: The work is somewhat relevant to the Web and to the track, and is of narrow interest to a sub-community

---

### Official Review · Reviewer_CMVF · 2024-11-21

**Novelty:** 2
**Technical Quality:** 1

**Review:**

Summary：

This paper presents IllusionCAPTCHA, an innovative CAPTCHA system designed to be easily solvable by humans but challenging for AI, particularly Large Language Models (LLMs). Through empirical studies, the authors evaluate LLM performance on various CAPTCHA types and find that while LLMs struggle with complex reasoning CAPTCHAs, they improve with Chain-of-Thought prompting. IllusionCAPTCHA leverages visual illusions, which are intuitive for humans but confounding for AI, and employs structured questioning to mislead bots. The study reports an 86.95% success rate for human users on the first attempt, outperforming traditional CAPTCHAs. However, this research is not without its issues, calling into question the robustness of the findings and their applicability in diverse scenarios.

Strengths：

1. The research conducted in Section 4, "Empirical Study," on LLMs' ability to solve existing CAPTCHAs and the associated human perception surveys contributes meaningfully to the subsequent studies on CAPTCHA. This section offers valuable insights into the efficacy of current CAPTCHA mechanisms against LLMs and human users' interaction with them, thereby informing future developments in CAPTCHA research.

2. The structure of the entire manuscript is exceedingly clear, and the reading flow is remarkably smooth. This clarity in organization greatly facilitates comprehension, enhancing the academic rigor and accessibility of the content.

Weaknesses:

1. Section 4.1 presents a study on LLMs' ability to solve existing CAPTCHAs and the associated human perception surveys, which holds significant implications for subsequent CAPTCHA research. However, the authors have not disclosed specific details of the dataset, such as the quantity and types of various images, which is a critical omission. Additionally, the number of experimental trials was not specified. Given the potential randomness in LLMs' reasoning, conducting multiple trials on the same image would be a reasonable approach, yet this aspect is not addressed in the paper.

2. In Section 4.2 "User Study," the experiment involving CAPTCHAs, which is time-constrained and employs a questionnaire format, is somewhat limited by the small sample size of 23 participants. This modest number undermines the persuasiveness of the study's conclusions. Similarly, in Section 6.4, where the new CAPTCHA is tested, the involvement of only 23 participants is deemed insufficient for a robust experimental validation.

3. Section 4.2 "User Study" lacks ethical review, which is a critical component in research involving human subjects. The use of characters like Donald Duck (image) and Mickey Mouse (text) in Figures 5 and Section 5.1 may also raise copyright issues with Disney, as these are protected intellectual properties. It is imperative for academic research to adhere to ethical standards and respect copyright laws to avoid legal repercussions and maintain academic integrity。

4. The process outlined in Section 5.1, "Illusionary Image Generation," is somewhat succinct, lacking detailed theoretical underpinnings. For instance, the methodology for calculating the similarity between generated images and the base image is not elucidated, nor is it explained why images with lower similarity would be more challenging for LLMs to discern. This necessitates a more robust theoretical framework to support these assertions.

5. Figure 5 illustrates the generation of an illusionary image; however, the manuscript also introduces a method for embedding words within images. Presenting both techniques together would enhance comprehension for readers. This integration could provide a more holistic view of the illusionary image generation process, offering a clearer theoretical foundation for why images with lower similarity to the base are more challenging for LLMs to recognize.

6. The user interface for the critical CAPTCHA, including examples of image description options, lacks visual representation, which impedes a clear understanding of the specific implementation of IllusionCAPTCHA. The absence of illustrative figures makes it difficult for readers to grasp the nuances of the system's operation.

7. In Tables 4, 5, and 6, it would be beneficial to consolidate the data into a single table to more starkly contrast the differences in success rates between LLMs and human users. The success rates of 100% and 0% in Tables 4 and 6, respectively, appear too absolute and may be indicative of a limited number of trials, which does not align with the principles of randomness. A more comprehensive dataset would provide a more accurate representation of performance variability.

**Questions:**

1. Could the detailed data from the experiments involving the dataset, LLMs, and human participants be provided within the paper?
2. What is the interface of the CAPTCHA? Can a more intuitive demonstration be provided?

**Ethics Review Description:**

The authors conducted two human participant experiments with 23 participants each, but did not provide information regarding the approval from the Ethics Review.

**Ethics Review Flag:**

Yes

**Reviewer Confidence:**

4: The reviewer is certain that the evaluation is correct and very familiar with the relevant literature

**Scope:**

3: The work is somewhat relevant to the Web and to the track, and is of narrow interest to a sub-community

---

### Official Review · Reviewer_K3v2 · 2024-11-30

**Novelty:** 6
**Technical Quality:** 3

**Review:**

The paper covers a study on the performance of both LLMs, as well as humans, solving CAPTCHAs. The paper shows that LLMs are getting increasingly better at solving CAPTCHAs. Interestingly, the paper also demonstrates that humans struggle on more complex CAPTCHAs. As such, the paper proposed a new CAPTCHA technique that is designed to be harder for LLMs, yet easier for humans to solve.

Thank you for submitting your work. I think the idea behind this paper is very interesting. I especially liked that there are two angels to the study: 1) how good are LLMs at bypassing current techniques, 2) how bad are humans at solving CAPTCHAs. Combining both results yields strong arguments for a new CAPTCHA system, which the authors also provide. However, there is an overall lack of detail throughout the paper which unfortunately undermine the results. Concretely, there is no detail provided on the nature of the data, the survey used in the human subject study, the pipeline of the LLM study, and the image generation.

For the dataset, I would have liked to see how you obtained the data, how the CAPTCHAs were picked, and what the reasoning behind that was. Additionally, I would have liked to read how many entries each category contained, some examples of how they looked like, and ideally their prevalence in the wild (either included in the dataset or through a measurement study).

For the user study, the survey is missing from the paper. I would have liked to see what questions were asked, what the trends were in the user’s answers, and most importantly how the CAPTCHAs were presented to them. Furthermore, I miss some information on how the users were picked. Because this study involves human subjects, I flagged the paper for ethics. I feel the paper could elaborate more on its ethical aspects. The intro contains a short paragraph on this, but this is insufficient. I would also like to read whether an IRB was consulted or not, and what their comments were (or an explanation why it was not involved)

For the LLM study, there are no details provided on the setup. What prompts were used? How many prompts were used? How many attempts did the LLM get to solve the CAPTCHA? Did you explore different prompting techniques? The study uses two different models. While I like that multiple models are compared, I think the paper would benefit from evaluating more models on this type of task.

Lastly, the illusionary image generation sections lacks detail on this new approach. It is unclear how the images were chosen and how many there are used. The section that describes this technique (Section 5.1) is also hard to follow and I don’t feel I got the full details on how this method works.

Overall, I think this research is novel and should definitely be published. Unfortunately, I also think that in its current state it is not yet ready for publication. Nonetheless, if my above points are addressed, I see a clear path to acceptance. Please see below some more comments in bullet point format, as well as some nitpicks.

Comments

- Related Work is missing
- Section 2.2 could be expanded with more details on LLMs
- Section 4.1 contains a lot of overlap with Section 2
- Table 1 and Table 2 lack clarity. Table 2 says it concerns LLM performance, but the text mentions it provides results of the user study.
- you mention “some individuals needing more than five attempts to successfully pass them” this is surprisingly a lot. Could you give details on the users groups used for this study? For example, how (un)familiar are they with CAPTCHAs?
- ‘To ensure that human users can still recognize the text with minimal effort, we opt for simple, familiar English words such as "day" or "sun".’ —> doesn’t this also make it easier for the LLM to recognize it?
- “Compared to text-based CAPTCHAs, ours is more user-friendly, as it avoids the challenges posed by visual illusions.“ —> I am confused, isn’t your solution to actual use visual illusions?
- “These hints are crafted to trigger hallucinations in LLMs, further increasing the likelihood that bots will select incorrect responses.” —> why do they trigger hallucinations? Do you have a source or analysis that demonstrates that these actual cause hallucinations?
- Tables 4, 5, and 6 are again very confusing. For example, Table 4 claims 100% success rate, but Table 6 claims 0%, both for RQ2. Furthermore, the order of the tables need to match the order in which they are discussed in the text to improve clarity and readability.
- The image generation section needs way more detail

Nits

- Introduction, first paragraph, you wrote “traditinoal” instead of traditional
- Section 2.2, I cannot parse the second sentence “By increasing the scale of training data, model can significantly…”
- In 4.1 there is an empty citation ([]) on the top right
- Type in 4.1 in “(4) Noise-enhanced Overlapping Text-based CAPTCHAs, shown in Figure 1(d), are the type of challenges combine both…” —> should be “…challenges that combine both…”
- Section 4.3, first paragraph, you write “…extracting CAPTCHA images from the their original…” should be either “the” or “their”
- Figure 4 has a typo in the last step, it says CAPTHCA instead of CAPTCHA
- Section 5 is titled “Methodology” but it does not signify of what. This is very nitpicky, but I think it would improve clarity.
- First sentence in 5.3 “Building on the our empirical study,”
- Chain of Thought is inconsistently abbreviated throughout the text, both as “COT” and “CoT”

**Questions:**

- In Section 3 you mention that passing captcha can lead to unauthorized access. My understanding is that the captcha does not determine whether someone is authorized. Could you please clarify this?
- How many attempts per captcha did the LLMs get?
- You mention CAPTCHAs were considered frustrating to the users. Did you survey them about this? If so, please provide the details as these are interesting insights.
- In Figure 5, I do not really see “what” the illusion is. It seems like two pictures overlayed. Could you elaborate?
- For the Illusionary Image Generation, how many images were there? How were they chosen? Again, lacking detail
- If there is a fixed set of well-known images, wouldn’t a trivial attack be to provide this list to the LLM and let it pick among those options only?
- How are the other options of the multiple choice picked? How closely related to the actual answer are they?
- To evade the “long-answer choice” can’t you just prompt the LLM to not look for the longest description? Or even just hardcode a check for that (as an adversary)

**Ethics Review Description:**

I want to flag this just because the paper lacks some detail on the ethics and the study involves humans. While I do believe that no ethics were violated, I just want to make sure the authors elaborate on their ethical practices in the paper. To the very least, some details should be given on whether an IRB was involved or not, what their comments were, or why it was not involved.

**Ethics Review Flag:**

Yes

**Reviewer Confidence:**

3: The reviewer is confident but not certain that the evaluation is correct

**Scope:**

4: The work is relevant to the Web and to the track, and is of broad interest to the community

---

### Official Review · Reviewer_tgzX · 2024-12-03

**Novelty:** 4
**Technical Quality:** 3

**Review:**

## Strengths
+ First anti-VLM CAPTCHA
+ Approach is practical and can be quickly deployed on the web
+ Approach is proven effective against VLM agents

## Weaknesses
- Empirical study fails to highlight prevalence or impact of real-world VLM-based attacks
- The proposed solution is highly niche: it focuses on anti-VLM, which has potential vulnerabilities against other more common attack vectors
- User study lacks rigor: no timing, environment, and demographic information

I thank the authors for defining a project in this area. We’ve seen that VLM agents struggle with specific visual puzzles, and using subliminal cues and illusions against them is an interesting approach. At the same time, I have to say that the technical depth presented in this study is a bit too shallow in comparison to prior works, which raises concerns about its overall contribution to the web. I will detail them below.

## The user study lacks rigor
I am concerned that the information provided in Section 6.1 may not be sufficient to fully demonstrate IllusionCAPTCHA's contribution to the web ecosystem. My reasons are as follows: The study does not include any data on the distribution of time taken to solve the CAPTCHA, which can be helpful to see the general trend and also spot outliers. Besides, the study does not seem to emulate actual user environments. There is no information about the device (i.e., desktop, mobile) and browser, making it difficult to justify its use in production. Additionally, there is no demographic information about the users, making it difficult to determine whether the sample is representative of Internet users.

From my understanding, the reasons for failure (i.e., cultural factors) can be easily addressed. Instead of focusing on isolated cases, it would be more beneficial to investigate the underlying physiological and psychological causes of failure. This can be achieved through a large-scale, systematic study on diverse IllusionCAPTCHA challenges on Amazon Mechanical Turk or similar crowdsourcing platforms. All in all, I highly encourage the authors to refer to the methodology of existing CAPTCHA-related user studies. (Searles et. al., USENIX Security 23), (Gossweiler et al., WWW 09), (Bursztein et. al, IEEE S&P 11).

## The proposed approach is highly niche as an anti-bot solution.
Existing studies on in-the-wild CAPTCHA attacks have shown a continued reliance on human-driven solving farms (Nguyen et. al, IEEE S&P '24). Additionally, there are browser extensions and scripts, such as Buster, that leverage deep learning-based solvers. A highly specialized anti-VLM CAPTCHA may lack sufficient justification if it is vulnerable to conventional deep learning methods, which would diminish its overall significance. I recommend that the authors stage an attack on IllusionCAPTCHA with deep learning and computer vision solvers. This can yield valuable insights into its robustness in practice. A defense should not overfit and discriminate against its adversaries, otherwise it would become vulnerable through the path of least resistance.

Continuing from the point above, I remain skeptical about the prevalence of VLM-based CAPTCHA attacks in practice and whether they truly justify a different CAPTCHA design. Without strong empirical evidence demonstrating how common these attacks are and what kind of damage they might inflict, it’s hard to see the necessity for adoption, especially so when a part of the web has already adopted frictionless, proof-of-work, and fingerprinting CAPTCHAs like Cloudflare Turnstile. The empirical study in Section 4 could benefit from further development by focusing on the actual impact of VLM-based attacks on CAPTCHAs in real-world web applications and services.

Minor points:
I noticed that GeeTest has been using Stable Diffusion and ControlNet to enhance their image CAPTCHAs since 2023 (see https://blog.geetest.com/en/article/captcha-creation-meets-stable-diffusion). Have the authors made a comparison between IllusionCAPTCHA and GeeTest's enhanced CAPTCHAs? This could further support the novelty of this work.

**Questions:**

Could you provide more details about how the user study is conducted? Namely, how are participants selected, how is the test environment (e.g., devices, browsers) configured?

**Reviewer Confidence:**

4: The reviewer is certain that the evaluation is correct and very familiar with the relevant literature

**Scope:**

4: The work is relevant to the Web and to the track, and is of broad interest to the community

---

### Official Review · Reviewer_AahE · 2024-12-03

**Novelty:** 5
**Technical Quality:** 4

**Review:**

This paper studies the trustworthiness issue of CAPTCHAs when facing LLMs. The authors conduct an empirical study to evaluate the performance of MLLMs in solving CAPTCHAs and compare it with the recognition ability of human beings. Based on the concluded insights, this paper proposes the IllusionCAPTCHA to generate "HumanEasy but AI-Hard" CAPTCHAs.

Pros:
+ Good topic. The trustworthiness issues of and from large models are of interests and meaningful.
+ The empirical studies are worth noting. It makes this paper more solid.
+ The given solution to address the CAPTCHAs issue is effective.

Cons:
+ The writing of this paper should be improved.
+ RQ1 and RQ2 occur twice in this manuscript. I suggest making it sortable.
+ The experiments could be more solid.

**Questions:**

+ Could you please provide the effectiveness of the generated CAPTCHAs on traditional automatic CAPTCHAs recognition methods? I wonder if the generated CAPTCHAs are only effective to MLLMs.
+ Provide the user study results, or the conclusions will not be convincing. By this way, the user study scale should be expanded, 23 participates are not enough.
+ Adversarial examples could also mislead models to make wrong decisions, such as [1,2]. Why not compare the proposed IllusionCAPTCHA  with them?

[1]Towards adversarial attack on vision-language pre-training models
[2]Transferable multimodal attack on vision-language pre-training models

**Reviewer Confidence:**

4: The reviewer is certain that the evaluation is correct and very familiar with the relevant literature

**Scope:**

3: The work is somewhat relevant to the Web and to the track, and is of narrow interest to a sub-community